# Pharmacy Students’ Perceptions and Attitudes towards Online Education during COVID-19 Lockdown in Saudi Arabia

**DOI:** 10.3390/pharmacy9040169

**Published:** 2021-10-14

**Authors:** Saleh Alghamdi, Majid Ali

**Affiliations:** 1Department of Clinical Pharmacy, Faculty of Clinical Pharmacy, Al-Baha University, Al-Baha 1988, Saudi Arabia; saleh.alghamdi@bu.edu.sa; 2School of Life and Medical Sciences, University of Hertfordshire (Hosted by Global Academic Foundation), 6 (R5), Cairo 11835, Egypt

**Keywords:** pharmacy education, COVID-19, lockdown, online education

## Abstract

In March 2020, a national lockdown in Saudi Arabia due to the pandemic forced all educational institutions to complete their academic year via online education. This study aims to explore pharmacy students’ perceptions and assess their attitude towards online education during the lockdown. A cross-sectional self-administered survey was designed to collect responses of pharmacy students (from one college of pharmacy in Saudi Arabia) from December 2020 through January 2021. A total of 241 students completed the survey. Students’ responses indicated that they had easy access to the technology, online skills, motivation and overall favorable acceptance for online learning and examinations. There was a significant difference in the mean scores between the students from different years of study (*p* = 0.013) related to technology access, and the male students were in significantly more favor of online examinations than female students (*p* = 0.009). The majority of the students indicated that the lockdown had no or negative impact on their learning and training. Students have general acceptance for online education delivery due to more technology access and online skills. More research should explore the factors affecting and the extent of the impact of online education on student learning and training.

## 1. Introduction

The delivery of education to students who are not physically present with the help of satellite, video, audio, graphic, computer and multimedia technologies, is defined as distance education [1]. This mode of education delivery is not a new phenomenon and has been practiced in one form or another since the early 1900s [2]. In the higher education sector, while online learning has generally taken place through recorded lectures and online platforms before the pandemic, some universities postponed learning and teaching until further notice, due to the lack of information technology and necessary infrastructure for both students and teachers in the wake of the pandemic and ensuing lockdown. On the other hand, the majority of countries implemented distance or online education to best meet their learning outcomes. However, in high-income countries, the coverage of distance or online education is reported to be 80–85% whereas in low-income countries it is reported to be 50% [3]. Questions also remain about how to harmonize semesters and academic calendars, as some programs have been successfully implemented online, while others have not.

While there is a long and well-established history of studying the efficacy of teaching and learning at a distance, the research outcomes are controversial. By 1977, however, there was an agreement among researchers that whether a student learns more utilizing one medium or the other is likely to depend on how the medium is used and which medium is used [4]. Several studies that compare cognitive factors such as academic performance, achievement, examination results and grades in distance learning, in general, found no differences regarding the cognitive factors [5,6,7,8,9,10,11,12,13].

Research on other factors such as student satisfaction with the course has yielded more mixed results. Davis [14], Ritchie and Newby [15], and Vamosi, Pierce and Slotkin [16] found that distance-learning students were less satisfied with their distance learning classes than the students in traditional classrooms. On the other hand, there is some evidence which indicates that faculty and students show more favorable attitudes towards teaching and learning through distance learning once they had experienced teaching a course or taking a course in a distance learning format [13,17,18,19].

Pharmacy education in Saudi Arabia has gone through several evolutionary stages since 1959. Prior to 2002, King Saud University (KSU) was the only university in the Kingdom that offered a pharmacy degree. A four-year Bachelor of Pharmaceutical Sciences program started in 1959, which progressed to being a five-year program by 1979 with the introduction of clinical pharmacy discipline to the curriculum. By 2010, the five-year program was renamed as Bachelor of Pharmacy (BPharm), and a six-year Doctor of Pharmacy (PharmD) was introduced [20]. Both curricula contain classroom teaching, laboratory and tutorial sessions, as well as a spiral of experiential training, and both qualify graduates to be practicing pharmacists following achieving a pass mark in the national pharmacy license exam. In March 2020, a national lockdown in Saudi Arabia due to the pandemic forced all educational institutions to complete their academic year via a distance learning mode, of which online delivery was the main component. Our university (Albaha University) developed a contingency plan to adapt to the distance learning mode which included extra information technology (IT) support for the faculty and the students. As part of this plan, the teaching staff and the training preceptors were required to revise their teaching and assessment plan while ensuring that the learning outcomes were not compromised and maintaining the academic integrity of the online assessments. This plan also emphasized more comprehensive use of Rafid, which is a locally developed learning management system (LMS) used in our university. This study aims to explore the pharmacy students’ perceptions and assess their attitude towards the shift in the education delivery mode during the lockdown.

## 2. Materials and Methods

### 2.1. Design of the Study

A cross-sectional self-administered survey was designed to collect the students’ responses.

### 2.2. Instrument

The questionnaire was developed based on a relevant review of the literature. Some questions were newly developed to meet the aim of this research. The questionnaire consisted of three main parts. Part A consisted of demographic questions such as students’ gender, year of study and current Grade Point Average (GPA). Part B comprised multiple statements encompassing five domains: 1. Technology access (four statements); 2. online skills (11 statements); 3. motivation (six statements); 4. online versus face-to-face learning (four statements) and 5. online versus face-to-face examinations (five statements). Students’ responses on each statement were scored to assess their attitude and perception using a 5-point Likert scale. The 5-point Likert scale used in each domain was different as appropriate for the statements in that domain. For domain 1: no access at all = 1; very difficult = 2; difficult = 3; easy = 4; very easy = 5. For domain 2: did not use it = 1; always faced a problem = 2; often = 3; few times = 4; never faced a problem = 5. For domain 3: did not use it = 1; strongly disagree = 2; disagree = 3; agree = 4; strongly agree = 5. For domain 4 and 5: strongly disagree = 1; disagree = 2; neutral = 3; agree = 4; strongly agree = 5. Higher scores represented students’ positive attitude in each domain and vice versa. Part C collected students’ views regarding the advantages and disadvantages of online learning during the pandemic (multiple response questions), the impact of e-learning on their overall training and learning (multiple choice questions) and any training required for using online technology (multiple response questions). The questionnaire was developed and administered in English and Arabic languages. The questionnaire was piloted with five students and no amendments were required following the piloting. 

### 2.3. Validity and Reliability of the Instrument

The following steps were taken to ensure the validity of the questionnaire (all three parts of the questionnaire including the five domains in Part B of the questionnaire individually): I.Face validity: the questionnaire statements were checked by the researchers and two other expert academics to ensure their relevance, reasonability and that no ambiguity existed.II.IContent validity: the researchers and the two expert academics also checked the content of the questionnaire to ensure that the content of the instrument was logical and easy to understand.

Reliability analysis of Part B of the questionnaire revealed a Cronbach’s alpha value of 0.858, which indicates strong internal consistency. Good internal consistency was also demonstrated by each of the five domains individually (Cronbach’s alpha values: technology access 0.762; online skills 0.830; 0.752; online versus face-to-face learning 0.652; online versus face-to-face examination 0.831).

### 2.4. Sampling and Sample Size

The survey was intended to be administered to all pharmacy students in our college. The sample size was determined using an online SurveyMonkey^®^ sample size calculator. Based on the total number of pharmacy students (*n* = 312) and keeping the confidence level 95% and margin of error 5%, a sample size of 173 students was required.

### 2.5. Inclusion and Exclusion Criteria

All PharmD students at the Faculty of Clinical Pharmacy in Albaha University were eligible to respond to the survey. No other students were allowed to participate in this survey.

### 2.6. Distribution Method and Data Collection Period

The questionnaire was administered to the eligible students via an online link using SurveyMonkey^®^. The data were collected from December 2020 through January 2021.

### 2.7. Statistical Analysis

Data were downloaded from SurveyMonkey^®^ as Excel and SPSS files for analysis. Descriptive and inferential statistical analyses were conducted using SPSS (Version 24; IBM, Armonk, NY, USA). The descriptive analysis illustrated students’ demographic characteristics and responses in terms of frequencies, percentages and means with standard deviations. Furthermore, Mann–Whitney U test and Kruskal–Wallis statistical tests were employed to determine the effect of independent variables (gender, year of study, GPA) on dependent variables (score of each of the five domains and total score of all the five domains in Part B of the questionnaire).

### 2.8. Ethical Considerations

Ethical approval for this study was obtained from the Institutional Review Board (IRB) of Albaha University (approval number 43,100,686). The survey introduction informed the students about their voluntary participation, anonymity and confidentiality of the collected data and their right to withdraw their information at any time.

## 3. Results

### 3.1. Demographics

Out of 312 students, 241 completed the online questionnaire (response rate 77%). Of the respondents, 79% were male students. The majority of the students were from third year, followed by fourth-year and fifth-year students, respectively. The majority of the respondents had GPAs between 3.5 and 4. The demographics of respondents are presented in Table 1.

### 3.2. Technology Access

Most of the respondents reported the access to technology as easy or very easy. The mean score of the four statements in this domain ranged from 3.4 to 4.2 (max: 5) (Table 2). The mean score of the overall technology access domain was found to be 14.7 (±3.2) out of the maximum of 20 with no significant difference between the mean scores of male and female students and between the students with different ranges of GPA. However, a significant difference in mean scores was found between students from different years of study (*p* = 0.013) (Table 8).

### 3.3. Online Skills

The majority of the participants reported having never faced a problem or faced fewer times regarding their online skills. The mean score of the 11 statements in this domain ranged from 3.7 to 4.3 (max: 5) (Table 3). The mean score of the overall online skills domain was found to be 43.7 (±8.0) out of the maximum of 55. Although male students’ mean score in this domain was higher than female students and first-year students reported higher online skills mean scores compared to the students from the other years, no significant difference between the mean scores of male and female students, students from different years of study and students with a different range of GPA was found. 

### 3.4. Motivation

Most of the respondents agreed or strongly agreed with the statements in this domain. The mean score of the six statements in this domain ranged from 3.5 to 4.3 (max: 5) (Table 4). The mean score of the overall motivation domain was found to be 23.7 (±3.8) out of the maximum of 30 with no significant difference between the mean scores of male and female students, students from different years of study and students with different ranges of GPA. 

### 3.5. Online versus Face-to-Face Learning

The majority of the participants agreed or strongly agreed to the statements in favor of online learning except that the majority of them disagreed that they learned more in online education. The mean score of the four statements in this domain ranged from 3.0 to 3.7 (max: 5) (Table 5). The overall mean score of all the statements in this domain was found to be 13.2 (±3.7) out of the maximum of 20. Although internship students scored least and fifth-year students scored most compared to the students from other years in the favor of online learning, no significant difference between the mean scores of male and female students, students from different years of study and students with different ranges of GPA was found. 

### 3.6. Online versus Face-to-Face Examination

The majority of the participants agreed or strongly agreed to the statements in the favor of online examination as compared to face-to-face examination except one statement in this domain which was regarding examination being difficult because of the lack of presence and ability of staff to solve emerging problems. This statement was reverse phrased as compared to the other statements in this domain and we recoded it (i.e., strongly disagree being 5 and strongly agree being 1) for the purpose of further analysis. After this recoding, the mean score of this new statement was found to be 3.3 (±1.4). The mean score of the five statements in this domain ranged from 3.3 to 3.9 (max: 5) (Table 6). The overall mean score of all the statements in this domain was found to be 18.1 (±5.3) out of the maximum of 25. The mean score of male students (18.6) was found to be significantly higher than that of female students (16.5) (*p* = 0.009) (Table 7). No significant difference between the mean scores of students from different years of study and students with different ranges of GPA was found. 

### 3.7. Differences in Total Scores of All the Domains between Demographic Characteristics

A significant difference in the mean total score of all the domains was found between male and female students (*p* = 0.031) (Table 7), and between students from different years (*p* = 0.030) (Table 8). Post hoc analysis revealed that the internship students scored lowest whereas the first-year students scored highest. No significant difference in the mean total score of all the domains was found between students with different GPA ranges (*p* = 0.901) (Table 9).

### 3.8. Miscellaneous

Table 10 presents the students’ responses to miscellaneous items related to online education during the pandemic. The majority of the students were of the opinion that they needed extra training regarding time management related to online technology. The majority of the students agreed that e-learning through Rafid was beneficial in learning from home, saving time and being favorable for people with restricted mobility. The disadvantages which the majority of the students agreed to were having no direct interaction with instructors and long working hours on the computer being harmful. Interestingly, the majority of the students said that there was no or negative overall impact of e-learning on the students through Rafid. However, the majority of the internship students said that there was a negative or very negative impact of the pandemic and the subsequent shift in internship education program on their training.

## 4. Discussion

The current study illustrates the experiences of pharmacy students under the COVID-19 lockdown, and the impact of the lockdown on their learning from their perspective. This study includes the responses of the students from the first year of their pharmacy degree through the internship year (experiential learning) and therefore provides a wide range of experiences and opinions. The students scored higher in the technology domain, which iterates their easy access to technology. The majority of the students were found to be using mobile technology for their learning. Access to technology plays a crucial role in this modern age for distance learning, which is mainly conducted via online learning. The COVID-19 lockdown has highlighted the essential use of technology more than ever. Students of this age are already equipped with required technology skills and this facilitated online education to a greater extent [21,22]. Furthermore, Ali and colleagues reported that the pharmacy students in Saudi Arabia under the COVID-19 lockdown period utilized technology for their online learning, online lectures, accessing the resources, attempting the online examinations and thus in overall online learning. Therefore, our finding that the students had easy access to technology can be extrapolated to the assumption that this facilitated their online learning during the lockdown.

Similarly, the students in our study scored higher in the online skills domain, which resonates with the above discussion. The majority of the statements in this domain were focused on how the LSM (Rafid) in our college of pharmacy facilitated the online learning of the students. Every LMS has its own advantages and disadvantages. Since there was a sudden shift from in-person learning to online learning as the lockdown was imposed at very short notice, there was a heavy reliance on LMS. For this, technical support for students and staff is imperative as reported by Almetwazi and his colleagues for their college of pharmacy in Saudi Arabia [23]. Troubleshooting for faculty staff for the efficient use of LMS has also been emphasized on a global scale [24]. Our students scored higher in this domain, and this is the reflection of the extra efforts made by our IT department in providing support to our students and the faculty staff. Although the majority of the students in our study reported greater online skills, there were some students who scored low on this scale. It is pedagogically ethical to ask such students to blow a whistle for help when they need it.

The students in our study were found to be highly motivated, as reflected by their scores in this domain. Since technology plays a role in engaging and motivating the students [24,25], their high motivation can be attributed to their easier access to technology and higher online skills as discussed above. Moreover, it was notable that the students from all the years, male or female, and with a wide range of GPA were equally motivated, and this again reflects the efforts of our IT staff and faculty staff in training and engaging the students to facilitate their online learning, particularly via LMS.

When compared online learning and examination with face-to-face learning and examination, the students scored relatively lower (but still high) as compared to the other domains. Interestingly, although the students reported that they learned less via online mode, they preferred this mode. This can be explained by their responses in the same domain that they found online mode more comfortable as compared to the face-to-face mode. This finding is reiterated by other studies in which the students highlighted the advantages of online learning during the lockdown period such as the convenience of not having been required to travel to the campus, thereby saving time and money [24,26]. Along the same lines, the students found the online examination less stressful, as they were present in the comfort of their homes. This also made them able to concentrate on the examination more as compared to face-to-face examination. One factor that might have also contributed to students’ satisfaction with the online examination was the lesser weightage given to the examination (20%) as compared to the 80% given to the other assessments in every course, as per the requirement of the Ministry of Education in Saudi Arabia for the semester affected by the lockdown period [23].

Furthermore, we found that the internship students scored lowest whereas the first-year students scored highest (overall mean score of all the four domains). This can be explained by the fact that the first-year students are new to the university environment and generally have low expectations. In contrast, the internship students generally and relatively expect more support from the university, especially regarding their experiential learning, which was most adversely affected during the lockdown period. One contributing factor could be that the experiential learning required regular contact and a relatively greater extent of communication between the students and the preceptors which, according to the students’ responses, was one of the disadvantages which the students faced during the lockdown period. Moreover, our preceptors were not ready for the sudden shift to the online mode. This has been a learning experience for the preceptors as well as the academic staff. At the time of writing, this pandemic is expected to last longer, affecting all walks of life. In the wake of this, one particular study with pharmacy students has suggested a ‘hybrid’ campus mode to transform learning in a ‘new normal’ era [27,28]. Another interesting finding was more than half of the students reporting that they needed more training on time management during the lockdown period. This is apparent as the daily routine of most individuals is affected when they are at home all the time due to the lockdown. However, this is an important learning point for the faculty staff to provide students with training focused on time management in such situations. 

One of the limitations of our findings is that they are based on the students’ responses from one particular educational institute. However, similar experiences have been reported from pharmacy students across the country in other studies [24,27]. Moreover, although our study reported from the students’ perspective that their learning was negatively impacted by the shift to the online mode during the lockdown period, further exploration is necessitated to demonstrate the validity of this finding. Since there was a modification in the assessment distribution marks in the courses, it was not appropriate to compare the students’ marks in the examinations with previous years, in order to investigate the impact on student learning by means other than their perspective. Moreover, we did not record the socio-economic characteristics of the students in our study as we believed that our students have consistent socio-economic characteristics such as they receive stipend from the university, live in their own residential places, possess the same educational background, do not have student loans and are not engaged in part-time or full-time work. However, studies in other countries must explore the impact of these characteristics on their students’ engagement in distance learning. Further studies should also explore the processes of maintaining academic integrity in online assessments in relation to the students’ perceptions regarding these processes. 

## 5. Conclusions

The lockdown period imposed with a short notice period caused the sudden shift in the learning mode. The students’ technology access and good online skills made the learning feasible for them. The students opined that the online examinations were stress-free, and therefore they were able to concentrate more as compared to the examinations on campus. However, further research should explore how online examinations affect student learning utilizing means other than student perspective. Moreover, the faculty staff should work with the preceptors to improve the experiential learning experience for the students ensuring pedagogical outcomes. This should lead to further research to explore the affecting factors and determine the impact of the shift to online mode on student experiential learning and training.

## Figures and Tables

**Table 1 pharmacy-09-00169-t001:** Demographic characteristics of the respondents.

Gender	Number (%)
Male	190 (78.8)
Female	51 (21.2)
Year of study	
1st year	33 (13.7)
2nd year	36 (14.9)
3rd year	50 (20.7)
4th year	44 (18.2)
5th year	50 (16.6)
Bridging	15 (6.2)
Internship year	23 (9.5)
GPA	
3.5–4	99 (41.1)
3–3.49	28 (11.6)
2.5–2.99	41 (17.0)
2–2.49	53 (22.0)
Below 2	20 (8.3)

**Table 2 pharmacy-09-00169-t002:** Relative frequency distribution of students’ responses (*n* = 241).

DOMAIN: Technology Access						
Statement	No Access at All %	Very Difficult %	Difficult %	Easy %	Very Easy %	Mean (SD)
1. A fairly new computer (with high speed, large memory, speakers and webcam).	8.3	15.8	24.9	34.4	16.6	3.4 (1.2)
2. A computer with adequate software (latest version of Microsoft office, adobe acrobat, real player, internet explorer).	6.2	5.4	19.5	46.5	22.4	3.7 (1.1)
3. A fast internet connection at home.	5.4	12.9	33.6	32.4	15.8	3.4 (1.1)
4. Mobile technology (iPhone, iPad, smartphone).	1.2	1.7	10.4	49.0	37.8	4.2 (0.8)

**Table 3 pharmacy-09-00169-t003:** Relative frequency distribution of students’ responses (*n* = 241).

DOMAIN: Online Skills						
Statement	Did Not Use It %	Always Face a Problem %	Often %	Few Times %	Never Faced a Problem %	Mean (SD)
1. Finding information on the internet (using Rafid, search engines, web surfing).	2.1	7.5	20.3	45.6	24.5	3.8 (1.0)
2. Sending and receiving emails (announcement) with its file attachments via Rafid.	3.7	3.3	15.4	29.9	47.7	4.1 (1.0)
3. Downloading and/or uploading files to and from the website (Rafid).	4.6	4.1	12.9	28.2	50.2	4.2 (1.1)
4. Asking questions and making comments in online discussion or chat (Rafid).	10.8	4.1	9.1	23.7	52.3	4.0 (1.3)
5. Posting materials online (Rafid) such as texts or PowerPoint presentations.	17.4	2.9	13.7	26.1	39.8	3.7 (1.5)
6. Participating in an online voice conversation (Rafid chat).	10.0	7.9	12.9	28.2	41.1	3.8 (1.3)
7. Scheduling time to take timely online activity (Rafid).	7.1	13.7	17.4	29.5	32.4	3.7 (1.3)
8. English is a barrier to me when participating online through emails or discussions.	0.4	5.8	11.6	29.9	52.3	4.3 (1.0)
9. Participating in an online theoretical lecture via Rafid.	0.8	4.6	10.4	29.9	54.4	4.3 (0.9)
10. Participating in an online practical lecture via Rafid.	10.8	7.1	15.4	21.6	45.2	3.8 (1.4)
11. Attempting an online exam through Rafid.	0.8	9.5	16.2	40.2	33.2	4.0 (1.0)

**Table 4 pharmacy-09-00169-t004:** Relative frequency distribution of students’ responses (*n* = 241).

DOMAIN: Motivation						
Statement	Did Not Use It %	Strongly Disagree %	Disagree %	Agree %	Strongly Agree %	Mean (SD)
1. I am able to concentrate when reading long documents online.	1.2	9.5	28.6	47.7	12.9	3.6 (0.9)
2. I am willing to spend 10–20 h each week studying online.	0.8	10.0	17.0	46.5	25.7	3.9 (0.9)
3. I will take my friends’ advice regarding using online technology.	3.7	1.2	5.4	59.3	30.3	4.1 (0.9)
4. I will take my instructors’ advice regarding using online technology.	2.5	2.5	5.8	52.3	36.9	4.2 (0.8)
5. Quick technology and administrative support are important to my success in using online technology.	1.7	0.8	5.0	47.3	45.2	4.3 (0.8)
6. The efforts of technical support provided by online support technicians meet my needs/solve my issues.	14.9	6.2	11.2	45.2	22.4	3.5 (1.3)

**Table 5 pharmacy-09-00169-t005:** Relative frequency distribution of students’ responses (*n* = 241).

DOMAIN: Online versus Face-to-Face Learning						
Statement	Strongly Disagree %	Disagree %	Neutral %	Agree %	Strongly Agree %	Mean (SD)
1. I think I learn more in online education than in face-to-face education.	22.4	11.2	30.3	15.8	20.3	3.0 (1.4)
2. I prefer online education to face-to-face education.	17.8	14.1	17.8	18.3	32.0	3.3 (1.5)
3. I feel more comfortable participating in online course discussions than in face-to-face course discussions.	12.0	10.0	17.8	18.3	41.9	3.7 (1.4)
4. Online education requires more study time than face-to-face education.	14.9	16.6	24.1	22.4	22.0	3.2 (1.4)

**Table 6 pharmacy-09-00169-t006:** Relative frequency distribution of students’ responses (*n* = 241).

DOMAIN: Online versus Face-to-Face Examination						
Statement	Strongly Disagree %	Disagree %	Neutral %	Agree %	Strongly Agree %	Mean (SD)
1. Online examinations reduce stress and exam anxiety.	8.3	8.3	14.5	22.4	46.5	3.9 (1.3)
2. Online examinations allow students to focus and concentrate more on the questions.	11.6	12.4	10.8	21.6	43.6	3.7 (1.4)
3. Online examinations are easier in terms of attending them than face-to-face exams.	10.0	11.6	18.3	22.0	38.2	3.7 (1.4)
4. Online exams are fairer than face-to-face exams in terms of marking and fairness between students.	15.4	10.0	15.4	19.1	40.2	3.6 (1.5)
5. Online exams are difficult because of the lack of presence and ability of staff to solve emerging problems.	24.1	23.2	21.6	17.0	14.1	2.7 (1.4) *

* The statement was recoded, and the new mean was found to be 3.3 (±1.4).

**Table 7 pharmacy-09-00169-t007:** Comparison of total mean scores of domains between demographics characteristics (Mann–Whitney U test).

Domains	Gender	Mean (SD)	*p*-Value
Technology access	Male	14.7 (3.2)	0.774
	Female	14.6 (3.0)	
Online skills	Male	44.0 (8.3)	0.135
	Female	42.8 (6.7)	
Motivation	Male	23.9 (3.8)	0.191
	Female	22.9 (3.6)	
Online versus face-to-face learning	Male	13.3 (3.7)	0.492
	Female	12.8 (3.8)	
Online versus face-to-face examination	Male	18.6 (5.3)	0.009
	Female	16.5 (5.4)	
TOTAL	Male	114.4 (16.9)	0.031
	Female	109.6 (14.5)	

**Table 8 pharmacy-09-00169-t008:** Comparison of total mean scores of domains between demographics characteristics (Kruskal–Wallis test).

Domains	Year of Study	Mean (SD)	*p*-Value
Technology access	1st	15.7 (3.2)	0.013
	2nd	15.4 (3.0)	
	3rd	14.2 (3.3)	
	4th	15.0 (3.3)	
	5th	14.4 (3.0)	
	Bridging	14.7 (3.5)	
	Internship	13.3 (2.3)	
Online skills	1st	46.8 (6.6)	0.060
	2nd	43.8 (8.9)	
	3rd	41.4 (8.0)	
	4th	43.9 (8.4)	
	5th	44.2 (7.1)	
	Bridging	45.5 (8.9)	
	Internship	41.7 (7.8)	
Motivation	1st	24.4 (3.6)	0.173
	2nd	23.8 (4.3)	
	3rd	23.0 (4.4)	
	4th	23.6 (4.2)	
	5th	24.5 (3.0)	
	Bridging	23.1 (3.4)	
	Internship	22.8 (2.5)	
Online versus face-to-face learning	1st	13.2 (3.6)	0.077
	2nd	13.6 (3.9)	
	3rd	13.1 (4.0)	
	4th	13.3 (3.8)	
	5th	14.0 (3.0)	
	Bridging	13.8 (3.7)	
	Internship	11.0 (3.1)	
Online versus face-to-face examination	1st	19.0 (5.2)	0.403
	2nd	19.3 (4.9)	
	3rd	17.5 (5.4)	
	4th	16.9 (6.2)	
	5th	17.8 (5.0)	
	Bridging	19.1 (4.7)	
	Internship	17.4 (5.2)	
TOTAL	1st	119.2 (14.9)	0.030
	2nd	115.8 (16.9)	
	3rd	109.2 (15.6)	
	4th	112.6 (16.3)	
	5th	115.8 (13.5)	
	Bridging	116.2 (17.9)	
	Internship	106.2 (13.8)	

**Table 9 pharmacy-09-00169-t009:** Comparison of total mean scores of domains between demographics characteristics (Kruskal–Wallis test).

Domains	GPA	Mean (SD)	*p*-Value
Technology access	3.5–4	14.8 (3.2)	0.712
	3–3.49	14.8 (3.8)	
	2.5–2.99	14.8 (3.2)	
	2–2.49	14.8 (2.9)	
	Below 2	13.7 (2.9)	
Online skills	3.5–4	44.5 (7.3)	0.547
	3–3.49	43.8 (11.2)	
	2.5–2.99	42.6 (7.6)	
	2–2.49	43.5 (8.1)	
	Below 2	42.8 (7.1)	
Motivation	3.5–4	23.6 (3.8)	0.827
	3–3.49	23.3 (4.9)	
	2.5–2.99	24.4 (3.8)	
	2–2.49	23.5 (3.5)	
	Below 2	23.6 (3.3)	
Online versus face-to-face learning	3.5–4	12.7 (3.7)	0.320
	3–3.49	13.0 (4.5)	
	2.5–2.99	13.9 (3.3)	
	2–2.49	13.6 (3.8)	
	Below 2	13.3 (3.1)	
Online versus face-to-face examination	3.5–4	17.3 (5.6)	0.058
	3–3.49	18.1 (5.8)	
	2.5–2.99	17.6 (4.9)	
	2–2.49	19.6 (5.1)	
	Below 2	19.5 (4.4)	
TOTAL	3.5–4	112.9 (16.5)	0.901
	3–3.49	113.0 (22.1)	
	2.5–2.99	113.3 (16.0)	
	2–2.49	115.0 (14.7)	
	Below 2	112.8 (14.3)	

**Table 10 pharmacy-09-00169-t010:** Students’ responses to miscellaneous items related to online education during the pandemic.

Item	Number of Students Who Selected This Item (%)
Extra training needed in using online technology (MRQ)	
Using Rafid	63 (26.1)
Making an exam through Rafid	34 (14.1)
Computer skills	42 (17.4)
Typing and editing	39 (16.2)
Time management	128 (53.1)
Managing multi-media content	34 (14.1)
Using the web for education	85 (35.3)
Online communication skill	38 (15.8)
Advantages of e-learning through Rafid (MRQ)	
Learning from own home	188 (78.0)
Saves time	187 (77.6)
Favorable for people with restricted mobility	128 (53.1)
Disadvantages of e-learning through Rafid (MRQ)	
No direct interaction with instructors	116 (48.1)
No direct interaction among students	59 (24.5)
Costs of internet	91 (37.8
Working long hours on the computer can be harmful	131 (54.4)
The impact of e-learning on students learning (MCQ)	
Very negative	26 (10.8)
Negative	71 (29.5)
No impact	76 (31.5)
Positive	48 (19.9)
Very positive	20 (8.3)
The impact of COVID-19 pandemic and the subsequent shift in the internship educational program on your learning process (MCQ for internship students only)	
Very negative	6 (26.1)
Negative	9 (39.1)
No impact	6 (26.1)
Positive	2 (8.7)
Very positive	0 (0)

MRQ: Multiple response question. MCQ: Multiple choice question.

## Data Availability

The data presented in this study are available upon request from the corresponding author.

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
