# Peer review of "Pharmacy Students’ Perceptions and Attitudes towards Online Education during COVID-19 Lockdown in Saudi Arabia"

_pharmacy, 2021, doi:10.3390/pharmacy9040169_

Round 1
Reviewer 1 Report
The manuscript presents a relevant and timely topic. It applies a proposed evaluation tool for e-learning use and motivation in pharmaceutical education, which may be useful for other researchers.
Although it presents results from only one university and cannot make consistent inferences about broader institutional factors that may affect distance education, it provides useful information about student perceptions.
However, in order for readers to understand the setting in which the study takes place, two questions are fundamental:
- the description of the socio-economic conditions of the students
- the description of the online system adopted by the university and the educational processes developed during the pandemic.
This issues would be well discussed within the other results.
Author Response
Thank you for your time to review the manuscript. Please see our responses in the red colour font below.
The manuscript presents a relevant and timely topic. It applies a proposed evaluation tool for e-learning use and motivation in pharmaceutical education, which may be useful for other researchers.
Although it presents results from only one university and cannot make consistent inferences about broader institutional factors that may affect distance education, it provides useful information about student perceptions.
However, in order for readers to understand the setting in which the study takes place, two questions are fundamental:
- the description of the socio-economic conditions of the students. This issue has now been addressed in the limitations (at the end of the discussion section).
- the description of the online system adopted by the university and the educational processes developed during the pandemic. This information has now been added in the introduction section.
This issues would be well discussed within the other results.
Reviewer 2 Report
Thank you for your work on this manuscript. Student pharmacists have experienced a variety of challenges in learning throughout the pandemic. It is valuable to learn more about these challenges across a number of countries and regions.
There are significant revisions that need to be made before this manuscript can meet its intended objectives. An initial concern is that this research is limited to students at one college of pharmacy in one country. While the researchers appropriately report demographic data for the respondents, it is still difficult for readers to evaluate the degree to which the respondents are similar to or different from their own settings. This is needed for readers to be able to generalize the results onto their own experiences.
It's important that the researchers describe their LMS (Rafid) in greater detail in their Introduction and Methods. I am not familiar with Rafid, and only realized it was an LMS after I read the Discussion. Readers need to better understand how Rafid works so that they can make appropriate comparisons to the LMS that they use (e.g., Blackboard, Canvas).
In the Introduction, you make mention of past research on distance learning and the degree to which students prefer distance learning to more traditional methods. It's important that one be aware of the context in which these studies were performed (likely where students had a choice between distance and traditional modalities) and what was taking place under COVID restrictions, which in most countries resulted in students no longer having a choice regarding using distance learning...it was either learn online or not learn at all. Thus it may not be appropriate to place this study in the context of previous studies, which were performed under very different circumstances.
In the Methods, it is important to note that Likert-type scales DO NOT provide interval or ratio scale data, regardless of one assigning numeric values to various response levels. At best Likert-type scales provide ORDINAL scale data. As such, it is not appropriate to use parametric tests such as the t-test and ANOVA. This analysis should be re-performed using the non-parametric equivalents of the above tests (Mann-Whitney U and Kruskall-Wallis).
While the researchers provided some background regarding how the survey was pilot tested and achieved face validity, no assessment of the reliability and validity of the various subscales was reported. It is not appropriate to sum these scales and report an overall result unless this level of analysis is applied to the subscales. This is particularly concerning for Table 7, where comparisons are reported between men and women on these subscales.
Just what does it mean to be "in favor" of online examinations? In favor compared to what? (More "traditional" forms of examinations?) What steps were taken to assure academic integrity in the online examination process, particularly compared to steps that are taken in your traditional examination processes? And how did the respondents perceive the effectiveness of these steps in assuring the outcomes of the online exam processes.
The manuscript would benefit from English language grammar/usage editing.
Round 2
Reviewer 2 Report
Thanks for addressing my concerns, particularly with the statistical tests.